# The Development of a Multienzyme Isothermal Rapid Amplification Assay to Visually Detect Duck Hepatitis B Virus

**DOI:** 10.3390/vetsci11050191

**Published:** 2024-04-26

**Authors:** Shuqi Xu, Yuanzhuo Man, Xin Xu, Jun Ji, Yan Wang, Lunguang Yao, Qingmei Xie, Yingzuo Bi

**Affiliations:** 1Henan Provincial Engineering Laboratory of Insects Bioreactor, Henan Provincial Engineering and Technology Center of Health Products for Livestock and Poultry, Henan Provincial Engineering and Technology Center of Animal Disease Diagnosis and Integrated Control, Nanyang Normal University, Nanyang 473061, China; xushuqi0713@126.com (S.X.); myz3320200182@163.com (Y.M.); jzbetty@163.com (X.X.); wy15839925569@163.com (Y.W.); lunguangyao@163.com (L.Y.); 2College of Animal Science, South China Agricultural University, Guangzhou 510642, China; xqm0906@126.com (Q.X.); yingzuobi@163.com (Y.B.)

**Keywords:** duck hepatitis B virus, multienzyme isothermal rapid amplification, lateral flow dipstick, sensitivity, rapidity, visible detection

## Abstract

**Simple Summary:**

Duck hepatitis B virus was originally found in newly hatched Pekin ducks in the United States in 1980, which was followed by global spread and expansion to geese; thus, the virus may cause more financial damage to the poultry industry worldwide. In this study, an assay combining multienzyme isothermal rapid amplification (MIRA) and lateral flow dipstick (LFD) was developed for the efficient and rapid detection of DHBV. Followed by optimization testing of the reaction temperature and duration, the complete procedure for the newly developed MIRA assay for DHBV detection could be carried out by only using a water bath and only required 15 min. The newly developed MIRA assay was only confirmed to work with DHBV, had a 10 times higher sensitivity than the routine polymerase chain reaction (PCR) assay, and displayed better practicability for clinical samples. In line with these results, it was determined that the novel MIRA assay combined with LFD developed in this study would be a good alternative approach for DHBV detection in field clinics.

**Abstract:**

Duck hepatitis B virus (DHBV) is widely prevalent in global ducks and has been identified in Chinese geese with a high prevalence; the available detection techniques are time-consuming and require sophisticated equipment. In this study, an assay combining multienzyme isothermal rapid amplification (MIRA) and lateral flow dipstick (LFD) was developed for the efficient and rapid detection of DHBV. The primary reaction condition of the MIRA assay for DHBV detection was 10 min at 38 °C without a temperature cycler. Combined with the LFD assay, the complete procedure of the newly developed MIRA assay for DHBV detection required only 15 min, which is about one-fourth of the reaction time for routine polymerase chain reaction assay. And electrophoresis and gel imaging equipment were not required for detection and to read the results. Furthermore, the detection limit of MIRA was 45.6 copies per reaction, which is approximately 10 times lower than that of a routine polymerase chain reaction assay. The primer set and probe had much simpler designs than loop-mediated isothermal amplification, and they were only specific to DHBV, with no cross-reactivity with duck hepatitis A virus subtype 1 and duck hepatitis A virus subtype 3, goose parvovirus, duck enteritis virus, duck circovirus, or *Riemerella anatipestifer*. In this study, we offer a simple, fast, and accurate assay method to identify DHBV in clinical serum samples of ducks and geese, which would be suitable for widespread application in field clinics.

## 1. Introduction

Duck hepatitis B virus (DHBV), classified under the *Avihepadnavirus* genus of the *Hepadnaviridae* family with a genome comprising partly double-stranded DNA of 3021–3027 bp, replicates through the reverse transcription of RNA intermediates (viral pregenomic RNA) and coded viral polymerase proteins [1]. Highly similar to hepatitis B virus (HBV), the DHBV genome mainly contains three partially overlapping open reading frames (ORFs), which encode three major proteins: core antigen (binds to viral pre-genomic RNA and forms the nucleocapsid protein) and e-antigen (the secreted form of the viral core antigen), surface antigens (PreS/S proteins; forms the viral envelope), and viral polymerase (binds to the stem loop of pre-genomic RNA and activates the encapsidation) [2,3].

DHBV was originally found in newly hatched Pekin ducks with a high prevalence rate from commercial flocks in the United States in 1980, which was followed by global spread in recent years [1]. However, the vertical route is mostly used for DHBV transmission, and the subsequently persistent DHBV infection during duck breeding can severely impair duckling growth and egg hatching [4]. A recent study found that intrahepatic and extrahepatic DHBV infection deregulates glucose metabolism and decreases glucose tolerance, which may cause adverse effects on the growth and development of DHBV-infected birds and reduce their production performance [5]. Based on a DHBV investigation of serum samples from the Henan, Anhui, and Hubei provinces of China in 2017 to 2019, the proportions of DHBV-positive duck and goose flocks were 58.6% (17/29) and 22.7% (5/22), which suggest that there is a high prevalence and host range expansion of DHBV from ducks to geese [6]. Most notably, a few research studies found that changes in some amino acid sites in the pre-s protein of DHBV were related to higher pathogenicity, and the most representative one was the amino acid shift from glycine to glutamate at position 133 (G133E) in the pre-s protein of DHBV, which causes a temporary increase in viral proliferation and eventual hepatocyte death [7]. Through the artificial challenging test over 72 days, the livers of ducks infected with G133E DHBV demonstrated acute injury such as enhanced periportal cell growth, massive hepatocytes lysis, and apoptosis compared with the ducks that were infected with wild-type DHBV; additionally, there were greater losses of mean body weights in the birds infected with G133E DHBV compared with the birds with wild-type DHBV [7]. In another study previously reported by our group, a total of 540 serum samples were collected from 23 duck farms and 31 goose farms in Central and Eastern China from 2020 to 2022, and the positive rates in duck and goose flocks reached 52.2% and 35.5%, respectively [8]. And a genome sequence analysis suggested that the 11 obtained geese-originated DHBV strains were all of the G133E type, and 5 of the 12 duck-originated strains were DHBV variants harboring the G133E mutation site in the preS protein, which further indicates that DHBV with increased virulence and transmission ability was highly prevalent in Chinese ducks, and even geese [8]. Even though there were no remarkable clinical symptoms, these findings imply that the health of ducks and geese was also influenced by DHBV infection, and the global poultry industry may suffer financial losses due to DHBV infection. Considering the complex infection statuses of various pathogens in ducks and geese, an accurate and simple assay for DHBV screening is greatly required.

Different technologies have been used in China to identify DHBV, including virus isolation and other serological or molecular techniques. As a serological method, enzyme-linked immunosorbent assays (ELISAs) have been widely used for DHBV detection in field clinics [9]. However, the extensive use of ELISAs, ELISA-based assays, or other serological methods is restricted by the specificity related to binding between an antigen and antibody, their time-consuming nature, and intricate procedures [9]. Highly sensitive and specific molecular assays, including conventional polymerase chain reaction (PCR), quantitative real-time PCR, and other PCR-derived technologies have been widely used in inspection agencies or labs, but these methods always require expensive equipment and skilled operation [10,11,12]. These disadvantages, which mainly result from temperature cycling and amplicons analysis, constitute the insurmountable obstacle to their on-site application. Subsequently, isothermal amplification methods, such as loop-mediated isothermal amplification (LAMP), has been extensively used to identify different pathogens [10,13,14,15,16,17]. With the aid of pH indicators, the detection results of an LAMP assay for DHBV could be read with the naked eye [13]. However, it requires complex specific primer sets, and there is a greater risk of nonspecific amplification, which cannot be identified by agarose gel electrophoresis and color indicators [16,17]. Furthermore, complex specific primer sets are required for more binding regions in the DHBV genome; the genetic heterogeneity and complex recombination, as previously reported [6], also increase the risk of false negativity resulting from mismatches between various primer sets and genome sequences of variant DHBV strains. In recent years, molecular detection technologies related to recombinase polymerase amplification (RPA) and recombinase-aided amplification (RAA) have been developing rapidly and have been widely used due to the benefit of their isothermal amplification principle [18]. RPA and RAA assays have been applied in food safety, gender and variety identification, and, most importantly, the detection of various pathogens related to medical and animal diseases; rapid and sensitive detection methods for nucleic acids are necessary for the monitoring and clinical control of infectious diseases. The rapid development of this technology further promotes the application of isothermal nucleic acid detection methods in the field because they do not need high-temperature denaturation, annealing, or other steps; the degree of dependence on precision instruments is greatly reduced; and they are among of the most effective technical means to realize immediate detection in the field [19,20,21]. Furthermore, RAA and RPA assays could be performed using a temperature less than 40 °C, and the matching primer set or probe has a simple and easy design. These advantages make them more suitable for clinical detection than LAMP assays, which should be carried out at about 65 °C and need complex primer sets.

Recently, the innovative multienzyme isothermal rapid amplification (MIRA) method, circumventing technical problems with existing molecular techniques, has been developed and used to clinically detect several pathogens [22,23,24]. MIRA is a novel isothermal amplification method that does not require a temperature circler and can be completed in 10–30 min at 37 °C–40 °C [25]. Combined with a lateral flow dipstick (LFD), the interpretation of results can be completed with the naked eye without gel electrophoresis and imaging systems [25]. These advantages make this method very suitable for on-site detection in the field or in simple clinics without sophisticated equipment. In this study, to set up a detection approach that is more practical for field samples, a novel detection assay for DHBV based on an MIRA assay was developed and evaluated for its accuracy through technical testing and clinical samples.

## 2. Materials and Methods

### 2.1. Viruses and Clinical Samples

From 2020 to 2023, serum samples of 136 duck and 145 geese were collected from different farms in the Anhui (23 ducks from 4 flocks; 19 geese from 3 flocks), Hebei (27 ducks from 5 flocks; 24 geese from 4 flocks), Henan (31 ducks from 5 flocks; 33 geese from 6 flocks), Hubei (29 ducks from 5 flocks; 38 geese from 6 flocks), and Jiangsu (26 ducks from 5 flocks; 31 geese from 5 flocks) provinces, China for DHBV screening. The sera were isolated from the jugular veins of the ducks and geese by strictly following aseptic procedures with the approval of the South China Agricultural University Committee for Animal Experiments (approval ID: 2019-0136). Sampled sera were stored at −80 °C until use. Reference pathogen strains of duck hepatitis A virus subtype 1 (DHAV-1), duck hepatitis A virus subtype 3 (DHAV-3), goose parvovirus (GPV), duck enteritis virus (DEV), duck circovirus (DCV), and *Riemerella anatipestifer* (RA) were used for specificity testing in this study and were all maintained in our laboratory.

### 2.2. Genomic DNA Extraction

Following the manufacturer’s instructions, total viral DNA was extracted from 0.2 mL of each serum sample using the EasyPure Viral DNA/RNA Kit (TransGen Biotechnology, Inc., Beijing, China) in a separate room for nucleic acid extraction to prevent contamination. The extracted viral DNA was quantified (260/280 optical densityratios) using NanoDrop 2000 (Thermo Fisher Scientific, Waltham, MA, USA). The extracted DNA with OD260/OD280 of 1.8 ± 0.1 was considered to be eligible and stored at −80 °C until use.

### 2.3. Development of MIRA Assay for DHBV Detection

The recommended primer length for the MIRA assay, according to the manufacturers’ instructions, was 25–35 bp. After multiple alignments between different DHBV strains available from the GenBank database (supported by the National Center for Biotechnology Information), primer sets for MIRA assay and the matching nfo probe for integrated LFD (Weifang Amp-Future Biotechnology Co., Ltd., Shandong, China) detection were designed using a web server (https://www.ezassay.com/primer, accessed on 18 December 2023) (supported by Ezassay Biotechnology, Inc., Hong Kong, China) based on the conserved genome regions of Y220201 (duck-originated DHBV strain, accession no. OQ183601) and synthesized by Sangon Biotech Company (Shanghai, China). The reverse primers were labeled with biotin at their 5′ ends for attachment to the LFD. Based on the reaction principle of MIRA, the probe for the MIRA assay was about 45–55 bp. Tetrahydrofuran was tagged in the middle of an antigen marker (FAM) added to the probe’s 5′ end for nfo recognition, an internal nucleotide analog (dSpacer tetrahydrofuran residue) replacing a nucleotide was added at about 15 nucleotides from the 3′ end, and a C3 spacer (polymerase extension blocking group) was added at the 3′ end. Table 1 shows the information of all primer sets and probes of the MIRA assay for DHBV detection.

The MIRA reaction set for DHBV detection in this study was performed with a commercial kit (Weifang Amp-Future Biotechnology Co., Ltd., Shandong, China), which was developed from 29.4 µL of Buffer A, 2 µL of each 10 mM primer, 1 µL of template DNA, 2.5 µL of Buffer B, and 13.1 µL of double distilled water (ddH_2_O) to make up 50 µL reaction systems. For the MIRA assay combined with a lateral flow dipstick (MIRA-LFD) assay, 0.6 µL of 10 mM nfo probe was additionally needed. Following mixing with shaking, the MIRA or MIRA-LFD reaction systems were incubated in a water bath, respectively. The amplicons for the MIRA or MIRA-LFD assays were separated using agarose electrophoresis and analyzed using the Gel Doc system (BioRad, Hercules, CA, USA) via the Image Lab™ and Image J (version 4.1) software. After the MIRA test with the nfo probe, 5 µL of the reaction product was mixed with 95 µL of ddH_2_O and incubated for 5 min at room temperature. After that, 50 µL of the treated reaction product was added into the sample wells of the LFD system. The results were interpreted based on quality control and assay criteria. The amplicons were also examined using agarose gel electrophoresis. The reaction product of the MIRA assay and DNA extraction reagent (a 25:24:1 ratio of phenol, chloroform, and isoamyl alcohol manufactured by Solarbio biotech Co., Ltd. in Beijing, China) were mixed in a 1:1 ratio and separated by 2.0% agarose gel electrophoresis [21].

To acquire suitable reaction conditions for the newly developed MIRA assay for DHBV detection, a temperature optimization test was conducted with a range of 36 °C–40 °C. In addition, the optimum reaction time was evaluated by employing time points of 5, 10, 15, and 20 min.

### 2.4. Specificity and Comparative Sensitivity Analysis

After confirming the optimum reaction conditions, the established MIRA assay was performed to evaluate the specificity of the nucleic acids of DHBV and six reference pathogens, namely DHAV-1, DHAV-3, GPV, DEV, DCV, and RA. To determine the sensitivity of the newly developed MIRA assay for DHBV detection, 10-fold serial dilutions of a standard plasmid (pMD-18-T containing the entire genome of the Y220201 strain [p-Y220201]) ranging from 4.56 × 10^6^ to 10^1^ copies were used for detection and compared with the routine PCR assay as previously described [25]. The amplicons resulted from the specificity and comparative sensitivity test were all analyzed by agarose electrophoresis and LFD testing.

### 2.5. Detection Efficacy of Clinical Samples and Statistical Analysis

To establish the accuracy of the newly developed MIRA assay for DHBV detection, the detection rates of DHBV via routine PCR and MIRA assays for 34 preserved DHBV isolates were compared with serum samples of 136 ducks and 145 geese. SPSS 22.0 (IBM, Armonk, NY, USA) was used to calculate the *p* and kappa values for the detection results of the two assays.

## 3. Results

### 3.1. Screening for Primers and Probes

According to gel imaging, the target sequences were successfully amplified by both primer sets (Figure 1). However, the products of the DHBV-1 primer displayed brighter bands (126 bp) than those of DHBV-2 and without nonspecific amplification, which agreed with the LFD results. And the DHBV-1 primer displayed a nonspecific amplification band. Therefore, the DHBV-1 primer was employed with the probe for subsequent tests.

### 3.2. Optimization of MIRA Assay for DHBV Detection

To investigate the optimal reaction temperature, an MIRA assay was conducted for DHBV detection at various reaction temperatures (37 °C–40 °C), with the assay conducted at 38 °C resulting in the brightest band (Figure 2). Therefore, we adjusted the temperature of the assay to 38 °C. In the time optimization tests, the amplified band started to appear after 5 min and peaked at 10 min (Figure 3). Consequently, the optimum duration for the newly developed MIRA assay for DHBV detection was considered to be 10 min.

### 3.3. Specificity and Detection Limit of Newly Developed MIRA Assay for DHBV Detection

The amplified band only existed in the DHBV lane (Figure 4), and the test strip produced the same result. These results demonstrate that the newly developed MIRA assay for DHBV detection can specifically identify DHBV from DHAV-1, DHAV-3, GPV, DEV, DCV, and RA.

A standard plasmid (p-Y220201) with an initial copy number of 4.56 × 10^6^ was used to assess the detection limit of the MIRA. The electrophoresis results show that the MIRA assay could detect 45.6 copies and was compatible with the LFD readouts, but it was 10 times more sensitive than the PCR assay (Figure 5).

### 3.4. Clinical Sample Detection

To evaluate the accuracy and practicability of the newly developed MIRA assay for DHBV, the 34 DHBV isolates preserved in our lab and clinical samples consisting of 54 serum samples from ducks and 58 serum samples from geese were detected using the newly developed MIRA and the routine PCR assays. The developed MIRA assay for DHBV detected eight more samples than the routine PCR assay. A statistics analysis demonstrated a high agreement of DHBV detection between the newly developed MIRA assay and the routine PCR assay, indicating that the developed MIRA assay possesses good accuracy and practicability (Table 2).

## 4. Discussion

Because of the lack of distinct clinical symptoms of DHBV, the harm it causes to the health of infected ducks and the economic losses in the poultry industry caused by DHBV infection are often ignored by farm managers and researchers. The prevalence of DHBV in goose farms also revealed the host range expansion of DHBV, which raised more concerns [6]. Therefore, an effective, quick, and simple diagnostic tool is urgently needed. Currently, laboratory instruments are employed to detect DHBV using PCR, qPCR, and LAMP assays [13,18,26,27]. To overcome instrument restrictions, the MIRA-LFD assay was developed, which aimed to surpass PCR limitations, expand its application range, and decrease its time requirement.

The newly developed MIRA assay tool can simply and rapidly detect DHBV within 10 min and at 38 °C, and it only requires a constant temperature device, such as the water bath used in this study. Therefore, MIRA can be used for the analysis of several samples in a short time due to the lack of equipment restrictions. Notably, while the optimal reaction temperature of MIRA was 38 °C, the results obtained at 37 °C were also relatively decent. Referring to existing reports related to the recombinase polymerase amplification (RPA) assay combined with LFD, the developed RPA assay could be performed in a closed fist at body temperature to visibly detect canine parvovirus 2 [28]. In addition, a reverse transcription RPA assay was also performed at body temperature to detect foot-and-mouth [29] and canine distemper viruses [30]. Therefore, we infer that future MIRA technology development may take into account the optimal reaction temperature for the screening of primer pairs and facilitate the completion of the reaction at body temperature, thereby resulting in/providing a detection assay without equipment.

The newly developed MIRA assay method for DHBV detection only requires one pair of primer sets, which has a simpler design than the widely used LAMP assay. More importantly, the replication manner, persistent infection, and wide prevalence of DHBV caused the extensive genetic diversity and heterogeneity of the virus, including multiple site mutations and complex inner genotype and intra genotype recombinations [6,8]. In our previous report about a LAMP assay developed for DHBV detection, we compared it with SYBR green-dye-based quantitative PCR for DHBV detection in clinical samples [12,14]. However, the reported SYBR green-dye-based quantitative PCR method cannot detect about one-fourth of the clinical samples used in this study, which may be caused by a mismatch between the primer and mutated genome sequence of the recently variant DHBV strains according to the sequences of the strains with negative detection results [12]. Therefore, continuous updates of the primer and probe are necessary for amplification-based molecular methods. In this study, the MIRA assay using a simple primer set harbored a smaller risk of false negatives caused by mismatches between the primer sets and the changed target sequences. Meanwhile, the primer set was designed according to highly conserved regions of the DHBV genome. And the simple primer set design would also allow for easy primer updating if the mismatch exists in the future.

Using the nfo probe and LFD makes the detection results more convenient and further shortens the time for the detection procedure to be completed. The developed MIRA approach for DHBV detection could complete all detection steps in 15 min. Compared with conventional PCR, it saves time and can be observed with the naked eye via LFD, avoiding the disadvantage of conventional PCR products that require agarose electrophoresis and imaging systems to detect the amplicons [13,18,27]. The detection limit of the MIRA assay for DHBV detection was 10 times lower than that of the routine PCR assay. For clinical sample testing, MIRA identified eight more samples than the routine PCR assay, which was most likely a result of the MIRA assay’s high sensitivity. The sequencing results of the MIRA amplicon from the eight samples and the parallel positive sample for DHBV in the same flocks also supported this speculation. These advantages emphasize the ease of use of the MIRA-LFD assay and its potential for widespread application in field clinics and poultry farms. However, the MIRA-LFD assay still requires careful handling, limiting its use in resource-limited settings or field clinics without proper facilities. And a premixed reaction mixture could be supplied for practical use to simplify the preparation of reaction systems.

Notably, the potential for aerosol contamination arises from the high sensitivity of MIRA and the exposure of amplified products during or after the reaction. Using LFD can make the MIRA assay more convenient for DHBV detection but may also increase the risk of aerosol contamination. Consequently, to mitigate this risk, nucleic acid extraction, reaction mixture preparation, and an electrophoresis analysis should be conducted in distinct areas. Additionally, determining the amplicons without opening the tube cap would be safer; this is also the optimization direction for MIRA assays in the future. To eliminate the inconvenient and time-consuming process for template preparation, which comprises DNA extraction and purification, a research study about the developed RPA assay for African swine fever virus (ASFV) compared the detection results of ASFV DNA in 20 whole blood samples of pigs that were extracted using the Quick-gDNATM MiniPrep kit from ZYMO Research (Irvine, CA, USA) and a simple treatment approach (the samples were only mixed with 200 µL of lysis buffer from the commercial kit mentioned above at 70 °C for 20 min) [31]. The results confirmed that the developed RPA assay could simply detect the ASFV in clinical samples with a heating/lysis buffer approach and did not display a significant difference with the routine nucleic acid purification method with a silica gel adsorbent, which suggested that the RPA assay harbored a higher tolerance to DNA polymerase inhibitors compared with real-time PCR. Even though the reaction principle and system are similar, the MIRA assay used the novel recombinase with a different biological origin from that of the RPA assay, and further evaluation tests are required to determine whether it is tolerant to common amplification inhibitors.

## 5. Conclusions

In conclusion, a probe-based MIRA assay for DHBV detection was developed in this study to conquer the limitations of expensive equipment and longer detection times of the current available DHBV detection assays; this approach has been confirmed to be a highly sensitive and specific detection approach. Combined with the LFD, the newly developed MIRA assay for visible DHBV detection can be completed in 15 min, and it would be more appropriate for wide application in the clinical field for rapid DHBV detection.

## Figures and Tables

**Figure 1 vetsci-11-00191-f001:**
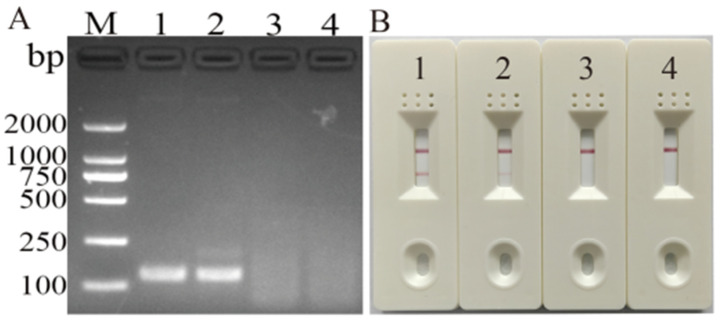
Screening analysis of primer pairs and probes. (**A**) Agarose electrophoresis of MIRA amplicons with probe and two MIRA primer pairs. M: DL 2000 marker; 1–2: amplicons resulting from DHBV-1 and DHBV-2 primer pair; 3–4: negative control for DHBV-1 and DHBV-2 primer pair. (**B**) LFD readout for MIRA amplicons with probe and two MIRA primer pairs; 1–2: amplicons resulting from DHBV-1 and DHBV-2 primer pair; 3–4: negative control for DHBV-1 and DHBV-2 primer pair.

**Figure 2 vetsci-11-00191-f002:**
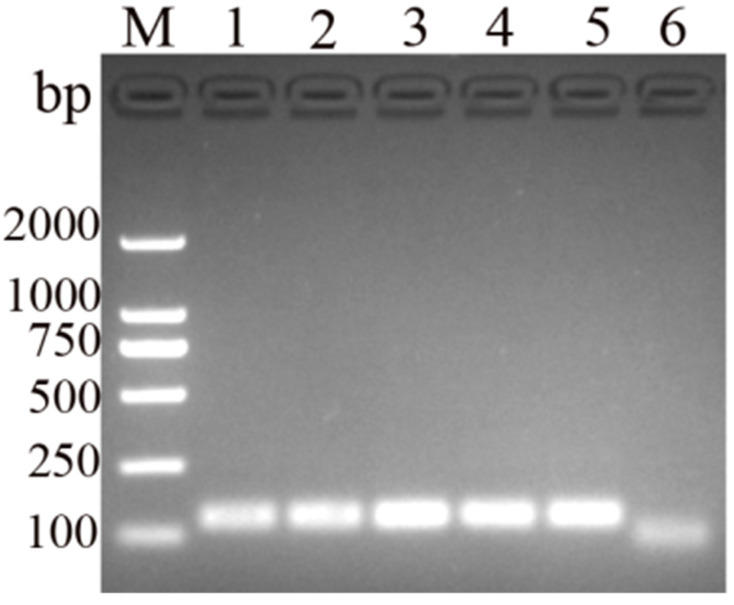
Amplicons of newly developed MIRA assays for DHBV detection at different temperatures. Lanes 1–5: 36 °C, 37 °C, 38 °C, 39 °C, and 40 °C, respectively; 6: negative control (band in this lane resulted from leftover reaction mixture product with primer–probe polymer).

**Figure 3 vetsci-11-00191-f003:**
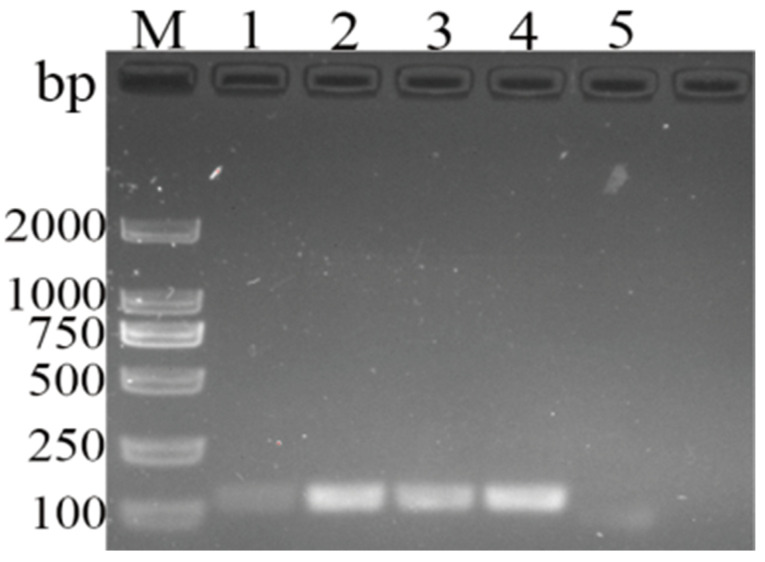
Amplicons of newly developed MIRA assay for DHBV detection at different reaction times. Lanes 1–4: 5 min, 10 min, 15 min, and 20 min, respectively; 5: negative control.

**Figure 4 vetsci-11-00191-f004:**
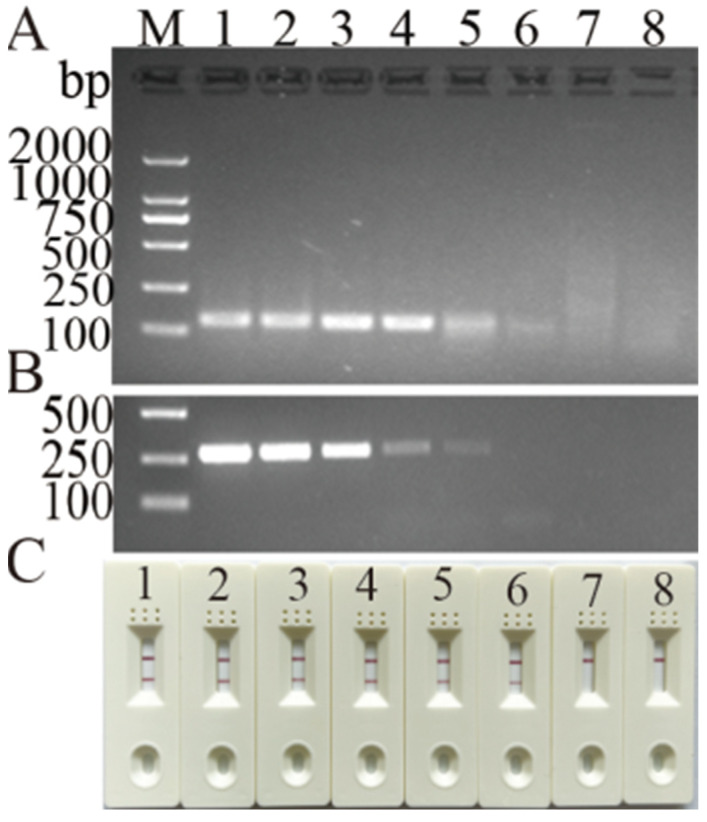
Detection limit of newly developed MIRA and routine PCR assays for DHBV. (**A**) Electrophoresis results of MIRA amplicons. (**B**) Routine PCR assay. (**C**) LFD readouts of MIRA amplicons. Lanes or strips 1–7: standard plasmid corresponding to 4.56 × 10^0^–10^6^ copies; 8: negative control.

**Figure 5 vetsci-11-00191-f005:**
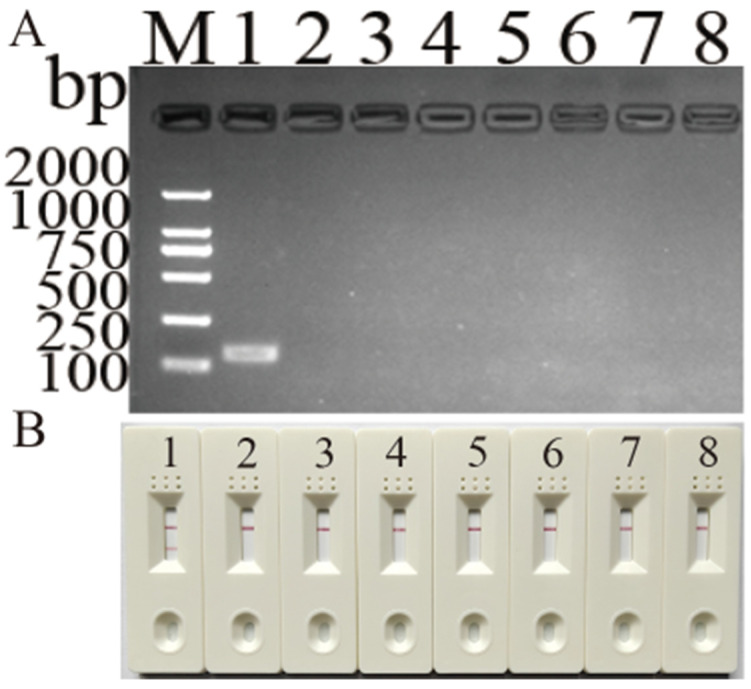
Specificity of newly developed MIRA assay for DHBV detection. (**A**) Electrophoresis results of MIRA amplicons; (**B**) LFD readouts of MIRA amplicons; 1: DHBV; 2: DHAV-1; 3: DHAV-3; 4: GPV; 5: DEV; 6: DCV; 7: RA; 8: negative control.

**Table 1 vetsci-11-00191-t001:** Information on primer pairs and probes of MIRA assay for DHBV detection.

Primer	Sequence (5′–3′)	Localization
DHBV-1F	G***M*^2^**GTCGGTCTCAGCCCTTTTCTCCTCCA	1619–1644 ^1^
DHBV-1R	[5′-Biotin]-ACGAGCGTTTGGGTGGCAGAGGAGGAAG	1717–1744
DHBV-2F	AAATA***Y***AATCCTGC***K***GACGGCCCATCCA	2423–2450
DHBV-2R	[5′-Biotin]-GGGGTGTAA***Y***T***Y***TTAAGTTCCACATAGC	2521–2548
Probe for DHBV detection	[5′-FAM]-TCTCTTCACTACTGCCCTCGGATCAGAAATC/IDSP/CTCGTCGCTTTAA-[3′ Spacer]	1645–1689

^1^ The localizations of the primer and probe are based on the genome sequence of the Chinese duck-originated DHBV strain of Y220201; its GenBank accession number is KY354038. ^2^ The letters in italic and bold represent degenerate oligonucleotides; M denotes A + C, Y denotes T + C, and K denotes for T + G.

**Table 2 vetsci-11-00191-t002:** Comparison of newly developed multienzyme isothermal rapid amplification and routine polymerase chain reaction assays for duck hepatitis B virus detection.

Samples	Detection Results		Statistics Analysis	
	Routine PCR Assay	MIRA Assay	Kappa (*k*)	*p*-Value of Kappa
DHBV isolation	34/34	34/34	1	<0.001
Serum from ducks	54/136	59/136	0.924	<0.001
Serum from geese	58/145	61/145	0.957	<0.001

DHBV, duck hepatitis B virus; PCR, polymerase chain reaction; MIRA, multienzyme isothermal rapid amplification.

## Data Availability

All of the data generated or analyzed during this study are included in this article, which are available from the corresponding author upon reasonable request.

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
