# Peer review of "The Development of a Multienzyme Isothermal Rapid Amplification Assay to Visually Detect Duck Hepatitis B Virus"

_vetsci, 2024, doi:10.3390/vetsci11050191_

Round 1
Reviewer 1 Report
Comments and Suggestions for Authors
The article “Simple and rapid assay for visibly detecting duck hepatitis B virus” by Xu et al, describes the development of a novel assay combining multienzyme isothermal rapid amplification (MIRA) and lateral flow dipstick (LFD) for rapid and efficient detection of duck hepatitis B virus (DHBV). The MIRA-LFD assay offers a simple, fast, and accurate method for identifying DHBV in clinical serum samples of ducks and geese, with potential widespread application in field clinics. Specific comment below:
-
Title: Consider revising the title to be more specific and informative. For example, "Development of a Multienzyme Isothermal Rapid Amplification Assay for Visually Detecting Duck Hepatitis B Virus."
-
Abstract: Clarify the novelty and significance of the study in the abstract. Specify how this assay improves upon existing methods. Include key quantitative results (e.g., sensitivity, specificity) in the abstract to provide a more comprehensive summary of the study.
-
Authors should provide a more concise overview of the prevalence and impact of DHBV in global duck and Chinese geese populations. Include specific statistics or figures to quantify the prevalence rates and economic losses associated with DHBV infection, if available. Additionally, including this statistic would underscore the importance of developing efficient and cost-effective diagnostic tools for managing DHBV infections in duck and geese populations.
-
2.1. Viruses and clinical samples- Provide more details on the collection process of serum samples, including the number of samples collected from each province and the sampling frequency. Mention any specific storage conditions or protocols followed for the serum samples.
-
2.2. Genomic DNA Extraction- Specify here the volume of serum used for DNA extraction and the concentration of the extracted DNA. In addition, also provide information on the quality control measures taken during DNA extraction and any specific precautions taken to prevent contamination during the extraction process.
-
Figure 1(A). Screening analysis of primer pairs and probes- There is a light band seen in the Lane 2 for DHBV-2 primer pair, Is it a nonspecific amplification band? And was this also the reason why authors chose DHBV-1 primer with the probe for subsequent tests.
-
Figure 2- There is a band in the Lane 6, Negative control- Is that a band or leftover reaction mixture product. Please specify either in result section or figure legend.
Discuss the practical implications of the MIRA-LFD assay for field clinics and poultry farms,
emphasizing its ease of use and potential for widespread application in context to
potential cost savings and efficiency gains associated with the adoption of the MIRA-LFD
assay.
8.
9. Authors should include the limitations of the MIRA-LFD assay such as risk of false negatives due to genetic diversity, necessitating continuous updates to primer and probe sequences. Additionally, while the assay reduces the need for expensive equipment, it still requires careful handling, limiting its use in resource-limited settings or field clinics without proper facilities. Further optimization and validation are necessary to address these limitations and ensure reliable performance.

Comments on the Quality of English LanguageThe quality of English in the article is generally good, but there are some areas where improvements could be made. There are instances of few grammatical errors for example, "The primary reaction condition of the MIRA assay for DHBV detection was 10 min at 38°C" could be "The primary reaction condition of the MIRA assay for DHBV detection is 10 min at 38°C". Overall, the article would benefit from some minor revisions for clarity and grammatical accuracy.
Author Response
Response to Reviewer 1 Comments
Point 1: Title: Consider revising the title to be more specific and informative. For example, "Development of a Multienzyme Isothermal Rapid Amplification Assay for Visually Detecting Duck Hepatitis B Virus."
Response 1: Thanks for your recognition. We have revised the title according to your advice.
Point 2: Abstract: Clarify the novelty and significance of the study in the abstract. Specify how this assay improves upon existing methods. Include key quantitative results (e.g., sensitivity, specificity) in the abstract to provide a more comprehensive summary of the study.
Response 2: Thanks for your advice. We have added the statements you mentioned in the abstract.
Point 3: Authors should provide a more concise overview of the prevalence and impact of DHBV in global duck and Chinese geese populations. Include specific statistics or figures to quantify the prevalence rates and economic losses associated with DHBV infection, if available. Additionally, including this statistic would underscore the importance of developing efficient and cost-effective diagnostic tools for managing DHBV infections in duck and geese populations
Response 3: Thanks for your advice. We have modified the introduction and added the statistics , for the research related to DHBV was very limited, we only added the statistics related to the Chinese reports in last years.
Point 4: 2.1. Viruses and clinical samples- Provide more details on the collection process of serum samples, including the number of samples collected from each province and the sampling frequency. Mention any specific storage conditions or protocols followed for the serum samples.
Response 4: Thanks for your advice. We have added the information related to collection , number of samples and specific storage conditions on the collection process of serum samples.
Point 5: 2.2. Genomic DNA Extraction- Specify here the volume of serum used for DNA extraction and the concentration of the extracted DNA. In addition, also provide information on the quality control measures taken during DNA extraction and any specific precautions taken to prevent contamination during the extraction process.
Response 5: Thanks for your advice. We have added the information you mentioned for the Genomic DNA Extraction.
Point 6: Figure 1(A). Screening analysis of primer pairs and probes- There is a light band seen in the Lane 2 for DHBV-2 primer pair, Is it a nonspecific amplification band? And was this also the reason why authors chose DHBV-1 primer with the probe for subsequent tests..
Response 6: Thanks for your advice. We have added the description related the nonspecific amplification band in Lane 2 in the results section.
Point 7: Figure 2- There is a band in the Lane 6, Negative control- Is that a band or leftover reaction mixture product. Please specify either in result section or figure legend..
Response 7: Thanks for your advice. The band in lane 6 was resulted from leftover reaction mixture product with primer-probe polymer. We have added the descriptions in the figure legend.
Point 8: Discuss the practical implications of the MIRA-LFD assay for field clinics and poultry farms, emphasizing its ease of use and potential for widespread application in context to potential cost savings and efficiency gains associated with the adoption of the MIRA-LFD assay.
Response 8: Thanks for your advice. We have discussed and added the statements for the practical implications of the MIRA-LFD assay for field clinics.
Point 9: Authors should include the limitations of the MIRA-LFD assay such as risk of false negatives due to genetic diversity, necessitating continuous updates to primer and probe sequences. Additionally, while the assay reduces the need for expensive equipment, it still requires careful handling, limiting its use in resource-limited settings or field clinics without proper facilities. Further optimization and validation are necessary to address these limitations and ensure reliable performance.
Response 6: Thanks for your advice. We have discussed the limitations of the MIRA-LFD assay and added the further optimization to address these limitations and ensure reliable performance.
Comments on the Quality of English Language
The quality of English in the article is generally good, but there are some areas where improvements could be made. There are instances of few grammatical errors for example, "The primary reaction condition of the MIRA assay for DHBV detection was 10 min at 38°C" could be "The primary reaction condition of the MIRA assay for DHBV detection is 10 min at 38°C". Overall, the article would benefit from some minor revisions for clarity and grammatical accuracy.
Response: Thanks for your advice. We have checked and improved the sentences.
Reviewer 2 Report
Comments and Suggestions for Authors
This is an excellent study that applied the MIRA method to develop a new detection method for DHBV. The methods and results are clear and easy to understand and represent the validity of the study. Some minor comments are noted below.
1. Table 1: Botin is Biotin?
2. In Section 2.3, The accession number or more information of the sequence used to design the primers for the MIRA method would need to be given.
3. Figure legend in Figure 4: 4.56 × 106–100 copies is 4.56 × 100–106.
4. In section 2.4, what was the reason for not using hepadnaviruses closely related DHEV, such as Orthohepadnavirus as control viruses to check the specificity of this method?
5. It is advisable to highlight where to look (where the positive bands appear) in the MIRA-LFD results in Fig.
Author Response
Response to Reviewer 2 Comments
Point 1: Table 1: Botin is Biotin?
Response 1: Thanks for the comments. We have changed Botin to Biotin.
Point 2: In Section 2.3, The accession number or more information of the sequence used to design the primers for the MIRA method would need to be given..
Response 2: Thanks for the comments. The accession number and the information of the sequence used to design the primers for the MIRA method was supplied in Table 1.
Point 3: Figure legend in Figure 4: 4.56 × 106–100 copies is 4.56 × 100–106..
Response 3: Thanks for the comments. We have corrected the figure legend according to your advice.
Point 4: In section 2.4, what was the reason for not using hepadnaviruses closely related DHEV, such as Orthohepadnavirus as control viruses to check the specificity of this method?
Response 4: Thanks for the comments. We have detected Orthohepadnavirus in duck and goose samples preciously, and no positive sample tested. Therefore, we have not using hepadnaviruses closely related DHBV as reference strain.
Point 5: It is advisable to highlight where to look (where the positive bands appear) in the MIRA-LFD results in Fig.
Response 5: Thanks for your comments. We have added the band size for the products of Primer Set 1 in section 3.1. We believed using arrow or other marks to highlight the positive band was not good for result displaying.